# A Simple Method to Establish Sufficiency and Stability in Meta-Analyses: With Application to Fine Particulate Matter Air Pollution and Preterm Birth

**DOI:** 10.3390/ijerph19042036

**Published:** 2022-02-11

**Authors:** Gavin Pereira

**Affiliations:** 1Curtin School of Population Health, Curtin University, Perth, WA 6102, Australia; gavin.f.pereira@curtin.edu.au; Tel.: +61-8-9266-3940; 2enAble Institute, Curtin University, Perth, WA 6102, Australia; 3Centre for Fertility and Health (CeFH), Norwegian Institute of Public Health, 0473 Oslo, Norway

**Keywords:** pregnancy, preterm birth, air pollution, meta-analysis, particulate matter

## Abstract

Fine particulate matter air pollution (PM_2.5_) is a potential cause of preterm birth. Inconsistent findings from observational studies have motivated researchers to conduct more studies, but some degree of study heterogeneity is inevitable. The consequence of this feedback is a burgeoning research effort that results in marginal gains. The aim of this study was to develop and apply a method to establish the sufficiency and stability of estimates of associations as they have been published over time. Cohort studies identified in a recent systematic review and meta-analysis on the association between preterm birth and whole-pregnancy exposure to PM_2.5_ were selected. The estimates of the cohort studies were pooled with cumulative meta-analysis, whereby a new meta-analysis was run for each new study published over time. The relative risks (RR) and 95% confidence interval (CI) limits needed for a new study to move the cumulative RR to 1.00 were calculated. Findings indicate that the cumulative relative risks (cRR) for PM_2.5_ (cRR 1.07, 95% CI 1.03, 1.12) converged in 2015 (RR 1.07, 95% CI 1.01, 1.14). To change conclusions to a null association, a new study would need to observe a protective RR of 0.93 (95% CI limit 1.02) with precision equivalent to that achieved by all past 24 cohort studies combined. Preterm birth is associated with elevated PM_2.5,_ and it is highly unlikely that any new observational study will alter this conclusion. Consequently, establishing whether an observational association exists is now less relevant an objective for future studies than characterising risk (magnitude, impact, pathways, populations and potential bias) and interventions. Sufficiency and stability can be effectively applied in meta-analyses and have the potential to reduce research waste.

## 1. Introduction

It is now well-accepted that fine particulate matter air pollution (PM_2.5_) is a cause of death from stroke, heart attack, lung cancer and chronic obstructive pulmonary disease [1,2,3]. These particulates are derived from heavy metals, sulphates and nitrates from incomplete fuel combustion, agricultural ammonia emissions, black carbon from burning diesel and biomass, sand, mineral dust and other specific sources that vary by geographic setting [4,5]. Due to its small size, PM_2.5_ can enter the blood stream via the lungs, reduce oxygen-carrying capacity, increase blood clotting and increase vasoconstriction [1]. The corresponding integrated scientific assessments are clear: the association between PM_2.5_ exposure and cardiovascular effects is “causal”, and associations with respiratory effects are “likely to be causal” [2]. Despite the establishment of plausible biological pathways, the scientific statement for effects of PM_2.5_ exposure on pregnancy and birth outcomes such as preterm birth is “suggestive, but insufficient to infer” causation, partially due to the lack of consistent observational epidemiological findings. Preterm birth, defined as birth before 37 weeks of gestation, is a major health problem and has a global incidence of approximately 10%, with approximately fifteen million cases born per year and a fatality rate of one million deaths per year [6]. Plausible biological pathways might involve preeclampsia, metabolic disease and fetal growth restriction, which are all risk factors for preterm birth, as well as mechanisms that promote preterm birth by increasing susceptibility to infection, interfere with placental development or trigger early activation of cytokines favoring inflammation [4]. Unlike the respiratory and cardiovascular endpoints for which causation has been concluded, underlying biological effects on pregnancy and the fetus are mediated by the mother and the placenta, and it is therefore reasonable to assume that such effects will be more difficult to detect. Variation in baseline maternal health and risk factors alone can contribute to inconsistent findings. Moreover, some degree of inconsistency among findings from observational studies is inevitable due to heterogeneity among populations in baseline levels of risk, exposure levels, and study methodologies. Nonetheless, a consequence of such inconsistency and heterogeneity is continuing observational research effort to identify *whether* PM_2.5_ exposure is associated with increased risk of preterm birth. It remains unclear as to whether a new study is needed to establish that an adverse association exists (*sufficiency*) and whether a new study is likely to change the aggregate evidence to date (*stability)* [7]. The aim of this study was to (i) develop a method to establish the sufficiency and stability of estimates of associations as they have been published over time, and (ii) apply the method to re-evaluate whether the observational evidence for an adverse association between preterm birth and whole-pregnancy exposure to PM_2.5_ is both sufficient and stable.

## 2. Methods

### 2.1. Design

This study was a cumulative meta-analysis, with the aim to ascertain the sufficiency and stability of observational associations between preterm birth and PM_2.5_.

### 2.2. Study Selection

The studies included were cohort studies on the association between preterm birth and PM_2.5_ that were selected in a recent systematic review and meta-analysis (SRMA) [8]. This SRMA was selected because it was the most recent (included peer-reviewed English-language articles published to October 2020) and employed restrictions to promote quality and minimise heterogeneity in design. There were 31 primary studies, all of which were cohort studies with Newcastle–Ottawa quality Scale scores ≥ 7. The need for the SRMA was conducted based on the premise that “preterm birth has been shown to be associated with prenatal air pollution exposure, but the results are still inconsistent” [8]. The authors of the SRMA reported that the pooled relative risk (RR) of preterm birth was 1.07 (95% CI, 1.05, 1.10) per 10 μg/m^3^ increase in whole-pregnancy exposure to PM_2.5_. The new methodology to ascertain sufficiency and stability derived in the present study is applied to the studies from the SRMA to show that this knowledge is not new, that estimates converged many years ago, and that it is unlikely that any new single study will change the aggregated evidence to date.

### 2.3. Outcome and Exposure Variables

Preterm birth was defined as birth before 37 completed obstetric weeks of gestation (≤36 + 6/7 weeks), which is derived either using the date of the first day of the last menstrual period or by ultrasound dating [9]. The focus of this study was on estimation of the total effect of whole-pregnancy exposure to PM_2.5_ on risk of all preterm births irrespective of clinical presentation, indication, and specific gestational timing prior to 37 weeks.

The focus of this study was on whole-pregnancy exposure to all-source mass concentration of particulate matter with aerodynamic diameter < 2.5 µm. Ideally, studies should derive whole-pregnancy exposure from conception, which is approximately two weeks after the first day of the last menstrual period, to the end of the 36th week (36 + 6/7 weeks) for both term and preterm births. For term births, this means that exposure between week 37 and birth are excluded because the pregnancy is not at risk of preterm birth during this period.

### 2.4. Data Extraction

Risk ratios (odds ratios, relative risks and hazard ratios) and their 95% CI limits were extracted for each primary study. When multiple estimates were available, the estimates corresponding to those with lowest expected exposure misclassification (i.e., spatially modelled vs. monitoring station measurement), those that were estimated for all-source exposure (i.e., typical baseline exposure vs air pollution event, all-source exposure vs. a specific anthropogenic source) and those that represented the whole target population (i.e., all pregnant women vs. sub-group based on biological susceptibility or sociodemographic vulnerability) were selected. Study-level characteristics of the total sample size, outcome prevalence, exposure distribution (mean, standard deviation, median, 25th centile, 75th centile and interquartile range) and year of publication/online release were extracted. Data were extracted by the author independently of the SRMA [8], then extracted in duplicate from the SRMA for validation. Minor differences were encountered. Differences included the following: the SRMA did not convert the odds ratios (OR) to RR (1 study); the SRMA misreported the preterm birth sample size (2 studies); and the SRMA used results for monitoring station exposures, but this study used results for modelled exposure (1 study).

### 2.5. Sufficiency of the Aggregate Evidence

Cumulative random effects meta-analysis was applied. The restricted maximum likelihood estimator was selected. To undertake the cumulative meta-analysis, the studies were ordered by increasing year of publication (the earlier of the year of availability online and the year of publication), and then meta-analysis was repeatedly applied after sequential inclusion of each newly published study over time. The observational evidence-base was considered *sufficient* at the time (year) at which no more additional studies were needed to establish an adverse association between PM_2.5_ and preterm birth. This definition was operationalised as the first completed year at which the lower limit of the 95% CI of the pooled RR exceeded 1.00 and the cRR differed from the final cRR by less than 0.01 per 10 µg/m^3^. However, researchers are free to choose another meaningful cRR difference and exposure contrast other than 0.01 and 10 µg/m^3^, respectfully. These were selected for this study because RR are often reported to a precision of two decimal places, and 10 µg/m^3^ was selected as the exposure contrast because the reviewed studies typically reported at least this much variability in whole-pregnancy exposure. Other possibilities for exposure contrast include reporting RR per standard deviation, or interquartile range increase in exposure.

### 2.6. Stability of the Aggregate Evidence

The observational evidence was considered *stable* if it is unlikely that an additional study would change the aggregate evidence. To achieve this, a *stability threshold* was estimated. If the relative risk of the current pooled estimate (RR_p_) is greater than 1, the stability threshold is defined as the maximum relative risk for a new study (RR_new_) to render the new pooled relative risk to 1 once this new study is added to the meta-analysis. Conversely, if the relative risk of the current pooled estimate (RR_p_) is less than 1, the stability threshold is defined as the minimum relative risk for a new study to render the new pooled relative risk to 1 once this new study is added to the meta-analysis. If the 100(1−α)% CI of RR_p_ contains 1, a new study is not needed to alter inference based on the interval estimate as it already includes the *null*. When the 100(1−α)% CI of RR_p_ does not contain 1, its confidence limit (RR_new.CL_) is defined as the relative risk for a new study to render the closest 100(1−α)% CI limit of RR_p_ to 1 after including the new study.

An extension of this formulation would be to define the stability threshold as the relative risk of a new study to render the new pooled relative risk to 1 ± δ, where δ is considered an acceptable threshold or tolerance.

The derivation of the stability threshold requires the current pooled estimate on the log scale (b_p_) and its variance (v_p_). If these are not available, they can be derived from the pooled relative risk (RR_p_) and its 100(1−α)% CI: (RR_lower_, RR_upper_) as
(1)$$b_{p}=\log\left(RR_{p}\right)$$
(2)$$v_{p}=\frac{\log\left(RR_{u}\right)-\log\left(RR_{l}\right)}{2\varphi^{-1}\left(1-\frac{\alpha }{2}\right)}$$
where $\varphi^{-1}()$ is the inverse of the cumulative distribution of the log(RR). With a standard normal distribution and α = 5%, the $\varphi^{-1}\left(1-\frac{\alpha }{2}\right)$ term is the familiar value of 1.96. After adding a new study, the new pooled estimate on the log scale is
(3)$$b_{p.new}=\frac{w_{new}b_{new}+w_{p}b_{p}}{w_{new}+w_{p}}$$
where w_new_ and w_p_ are weights applied to the estimates for the new study and the current pooled estimate, respectively. The stability threshold on the log scale is derived as the value of b_new_ such that b_p.new_ = 0, or equivalently, the relative risk of a new study (RR_new_) such that the pooled relative risk updated with this new study (RR_p.new_) is 1:(4)$$Stability\;Threshold=RR_{new}=exp\left(-\frac{w_{p}}{w_{new}}b_{p}\right)$$

The 100(1−α)% CI for the new pooled estimate on the log scale (b_p.new_) is $b_{p.\mathrm{new}}\pm \mathrm{SE}\left(b_{p.\mathrm{new}}\right)$, which can be expressed on the relative risk scale (RR_p.new_) as
(5)$$\exp\left(\frac{w_{new}b_{new}+w_{p}b_{p}}{w_{new}+w_{p}}\pm \varphi^{-1}\left(1-\frac{\alpha }{2}\right)\sqrt{\frac{w_{new}^{2}v_{new}+w_{p}^{2}v_{p}}{{\left(w_{new}+w_{p}\right)}^{2}}}\right)$$

The confidence limit for the stability threshold on the log scale (b_new.CL_) is derived as the value of b_new_ such that the closest 100(1−α)% CI limit to the null becomes 0 after pooling the new study, or equivalently, the relative risk of a new study such that the closest 100(1−α)% CI limit to the null becomes 1 after pooling the new study (RR_new.CL_). For a fixed effects meta-analysis,
(6)$$RR_{new.CL}=exp\left(-\delta \varphi^{-1}\left(1-\frac{\alpha }{2}\right)\frac{1}{w_{new}}\;\sqrt{w_{new}^{2}v_{new}+w_{p}^{2}v_{p}\;}-\frac{w_{p}}{w_{new}}b_{p}\right)$$
where δ is −1 and 1 if the lower and upper 100(1−α)% CI limits are closest to the null, respectively. Under inverse-variance weighting ($w_{\mathrm{new}}=v_{\mathrm{new}}^{-1},{\;w}_{p}=v_{p}^{-1}$), the formulae for the stability threshold and its confidence limit can be simplified, but $v_{\mathrm{new}}$ remains unobserved. Alternatives are to assume that $v_{\mathrm{new}}$ is the same as the current pooled estimate (*highly conservative*, selected for this study), the lowest variance among previous studies (*conservative*), highest variance among previous studies (*optimistic*) or a variance that is expected based on the prospective design of a new study (*designed*). Under the highly conservative scenario and inverse variance weighting,
(7)$$Stability\;Threshold=RR_{new}=exp\left(-\frac{v_{new}}{v_{p}}b_{p}\right)$$
(8)$$RR_{new.CL}=exp\left(-\delta \varphi^{-1}\left(1-\frac{\alpha }{2}\right)\sqrt{2v_{p}\;}-b_{p}\right)$$

The stability threshold and its confidence limit can be defined for random effects meta-analysis with estimators other than inverse-variance weights. Due to the large range of methods used to undertake meta-analyses, the stability threshold and its CI limit are estimated by computational simulation in this study. Specifically, a sequence of new study effect estimates are generated, included in a meta-analysis with the current pooled estimate, and the stability threshold and its confidence limit are defined as the first effect estimate that nullifies the new pooled estimate and its confidence interval limit, respectively.

### 2.7. Statistical Analysis

Cumulative meta-analysis was undertaken on the log relative risk scale and exponentiated back to relative risks. Consequently, estimates for studies that reported relative risks (RR) were left unchanged (2 studies), and estimates for studies that reported odds ratios (OR) were transformed to RRs (22 studies): $RR=\frac{OR}{1+p\left(OR-1\right)}$, where *p* was assumed to be the overall prevalence of preterm birth [10]. Due to the non-equivalence of relative risks and relative rates, cumulative meta-analysis was undertaken separately for time-to-event studies that reported hazard ratios (HR) (7 studies) [11]. The pooled HR were then descriptively compared to the pooled RR estimates.

Heterogeneity was assessed using I^2^. No specific threshold was used to define “high heterogeneity” a priori. Analyses were conducted in R version 4.0.4 (R Foundation for Statistical Computing, Vienna, Austria) with the *metafor* package [12].

### 2.8. Assessment of Bias

The restricted maximum likelihood (REML) estimator was used for the main analysis. However, the choice of estimator can affect the precision of the meta-analysis estimate and therefore also the interpretation of sufficiency and stability. To address this, all analyses were repeated as a fixed effects (FE) cumulative meta-analysis. Next, analyses were repeated with alternative choices for the random-effects estimator: DerSimonian–Laird (DL) estimator (inverse-variance weighted, least conservative) and the variance components (VC) estimator (equal-variance weighted, most conservative) [13,14].

A funnel plot was produced by plotting the standard error versus the log RR. A relationship between the observed effect estimates and their standard errors can result in asymmetry, which indicates potential for publication bias. A test of asymmetry was conducted using a regression test on the standard errors using the method proposed by Egger et al. [15].

### 2.9. Ethical Approval

The study was conducted in compliance with principles outlined in the Declaration of Helsinki. Ethical committee approval for this study was not required.

## 3. Results

The studies were published over a 12-year period from 2009 to 2012. There were 14 studies from the USA [16,17,18,19,20,21,22,23,24,25,26,27,28,29]; eight studies from China [6,30,31,32,33,34,35,36]; four studies from the European region [37,38,39,40]; two studies from Canada [41,42]; two studies from Australia [43,44]; and one study from Peru [45].

The prevalence of preterm birth and mean whole-pregnancy exposure to PM_2.5_ varied by geographic region (Table 1). The median prevalence of preterm birth among the studies was 7.80% (interquartile range (IQR) 5.70–9.05%). The lowest prevalence of preterm birth was 3.0%, which was reported for a study from Victoria, Australia [44], and the largest was 14.0%, which was reported for a study from Colorado, USA [27]. The median of whole-pregnancy mean exposure to PM_2.5_ reported among the studies was 13.6 µg/m^3^ (IQR 9.6 µg/m^3^–37.4 µg/m^3^). The lowest whole-pregnancy mean exposures were observed for the two studies from Australia (6.21 µg/m^3^ and 6.90 µg/m^3^), and the largest for studies from China (median 56.9 µg/m^3^, IQR 46.8 µg/m^3^–64.7 µg/m^3^).

By the end of 2015, the pooled estimates of the RR converged to 1.07 (95% CI 1.01, 1.14) (Figure 1). Observed estimates reported by studies published after 2015 resulted in negligible change to the point estimate and marginal improvements in precision. The cumulative relative risk at the end of 2020 was 1.07 (95% CI 1.03, 1.12) (Table 1). The stability threshold was 0.93 (95% CI limit 1.02) by the end of 2015 and remained unchanged by the end of 2020. Heterogeneity (I^2^) was 87% at the end of 2015 and remained high (89%) by the end of 2020.

Studies that reported hazard ratios were published over a short time period (2017–2019), which limited the ability to identify the time point of convergence of results (Table 2). However, results from the cumulative meta-analysis of hazard ratios yielded similar results to those for relative risks, with a final cumulative hazard ratio of 1.08 (95% CI 1.01, 1.14) and a final stability threshold of 0.93 (95% CI limit 1.02) and I^2^ of 98%.

Results pertaining to sufficiency were robust to the type of meta-analyses undertaken. When analyses were repeated using fixed effects, cumulative meta-analysis the cRR estimate was more precise (FE: cRR 1.06, 95% CI 1.04, 1.07) and converged by 2012 (FE: cRR 1.07, 95% CI 1.02, 1.12). Cumulative meta-analyses repeated with other random effects estimators produced less precise estimates of the cRR (DL: cRR 1.07, 95% CI 1.03, 1.12; and VC: cRR 1.08, 95% CI 1.02, 1.13). Applying the DL and VC estimators resulted in estimates that converged in 2015 and 2012, respectively (DL: cRR 1.07, 95% CI 1.01, 1.15; and VC: cRR 1.07, 95% CI 1.02, 1.12). Similarly, results pertaining to stability were also robust to the type of meta-analysis undertaken. Under fixed effects cumulative meta-anlaysis, the stability threshold was 0.93 (95% CI limit 1.02) for all estimators (FE, DL and VC) at the year of convergence, which was identical to that of the main analysis (REML). There was insufficient evidence for asymmetry based on the funnel plot (Appendix A) and Egger test.

## 4. Discussion

This study proposes a new method to identify whether a new epidemiological study is needed to ascertain whether an observational association exists between an exposure and an outcome. An empirical definition was made for the time at which current evidence can be considered *sufficient*. From this time, no more additional studies are needed to confirm the existence of an association based on the aggregate evidence. It was demonstrated that the cumulative evidence for the existence of an association between fine particulate matter and preterm birth has not changed since 2015, indicating that evidence for the existence of a statistical association was sufficient from that time. A method was presented to estimate the *stability threshold*, defined as the effect that a new study would need to alter inference. This study demonstrated that current statistical evidence is stable. That is, any new study would require an unusually precise *protective* effect to alter the conclusion of the existence of an adverse association between fine particulate matter and preterm birth. Results were robust to the type of meta-analysis (estimator). The implication of these findings is that whether an association exists in observational epidemiological studies is now less relevant an objective for future studies than characterising risk (magnitude, impact, pathways and populations), examining the influence of potential bias, and implementation of interventions.

Meta-analyses provide aggregate summaries of associations by pooling estimates across a range of studies into a single estimate. It is not uncommon that these pooled estimates are accompanied by high levels of statistical heterogeneity (I^2^), particularly for large meta-analyses of observational studies, which warrants further investigation of underlying aetiological differences [46]. Although some degree of heterogeneity is inevitable [47], high levels can prompt researchers to conclude that the effect estimate is uncertain and that further research is needed. The global increase in meta-analyses [48] might be partially attributable to such a phenomenon. For the association between fine particulate matter and preterm birth, there is strong observational evidence that an adverse association exists, and it appears that no further studies are needed to make this conclusion. The high heterogeneity reflects the wide dispersal of the individual study estimates, yet these estimates are somewhat consistently in the direction of an adverse association. That is, the directions of effects are stable, while their magnitudes are not. An investigation of the aetiological causes for the heterogeneity has not been undertaken. It is therefore plausible that pooled estimates cannot be interpreted because underlying causal effects are context-specific. Such context includes differences in particulate matter composition and sources, co-pollutant exposures, underlying biological susceptibility and socioeconomic vulnerability, and baseline prevalence of preterm birth and other infectious and chronic diseases in the study region [49]. Indeed, despite much higher concentrations in China, the relative risk estimates were similar to those observed for the US and were exceeded in the low-concentration setting of Australia (Appendix A).

The theory of sufficiency and stability of cumulative meta-analyses is not well-developed. However, the terminology was well-described and introduced to public health research in 2001 along with a metric to identify sufficiency [7,50]. This metric, the *failsafe number*, was developed in 1979 and was defined as number of unpublished studies (due to publication bias) with null results needed to render the pooled result statistically significant at the 5% level [51,52]. The *failsafe ratio* was then defined as the ratio of the failsafe number to a tolerance level of 5k + 10, where k is the number of studies in the meta-analysis. The failsafe ratio is based on the assumption that the number of unpublished null studies is five times the number of published studies and that there are at least 15 such unpublished studies. Failsafe ratios exceeding 1 would indicate that the meta-analysis is tolerant to future unpublished null results. The failsafe was developed to account for publication bias in a prospective indicator for sufficiency. In contrast, this study describes a retrospective indicator for sufficiency using confidence intervals as a proxy for statistical certainty and temporal changes in deviations of effect sizes from the final pooled effect size to identify convergence. The *cumulative slope*, obtained from sequential regressions of cumulative meta-anaysis effects on the number of studies over time, has been previously proposed as a measure of [50]. A cumulative slope that converges to close to zero would indicate temporal stability. In contrast, stability in this study was defined as the prospective stability of inference by estimating the magnitude and direction of effect needed for a future study to change our conclusions. It is possible that the cumulative evidence is temporally sufficient and stable according to the failsafe ratio and the cumulative slope but non-sufficient and unstable according to this study’s definition of sufficiency and the stability threshold. The methods proposed to identify sufficiency and stability in this study are not set within the hypothesis testing framework, are largely based on observed effect sizes rather than attainment of statistical significance and broadly conform to contemporary American Statistical Association recommendations [53].

Some limitations of this study are shared with the limitations of observational studies. Demonstration of sufficiency and stability of epidemiological findings does not imply that a biological causal association exists but does provide stopping criteria for replication, evidence to support a consensus epidemiological statement on the association and motivation to redirect research effort. Research effort should be directed to studies that employ the *precautionary principle of public health*, which would assume an association exists (e.g., impact assessments, intervention research). Future research efforts should also be directed to elucidate potential for systematic bias affecting the observed studies. The E-value for a meta-analysis can be defined as the minimum strength of association that an unmeasured confounder would need to have with both the exposure and the outcome to fully explain the pooled estimate [54]. The E-value for the pooled estimate from the meta-analysis is 1.34 (95% CI limit 1.21), which is similar to the magnitude of the association between poverty and preterm birth [55] and between poverty and particulate matter exposure [56]. Therefore, there is potential for systematic residual confounding by poverty. A related issue is that of potential selection bias from the lack of studies from regions such as south Asia and sub-Saharan Africa, which account for a large proportion of preterm births globally. Demonstration of sufficiency and stability of epidemiological findings does not imply that protective or null associations will not be observed by individual studies in the future but does quantify the size and direction of effect needed to alter our current conclusion. An inherent limitation of the method is that “convergence” is determined retrospectively having already observed the final pooled estimate, which means that it is possible for the pooled estimates to converge to the final estimate, temporarily diverge, and then converge again. However, if the final pooled estimate is considered the best estimate, then less emphasis should be placed on the studies that cause the transient divergence. A possible criticism of the proposed approach to establish stability is that it violates the contemporary American Statistical Association (ASA) recommendation to “[pay] no particular attention to whether the [confidence] interval includes the null value” [53]. Such criticism is unwarranted for the stability threshold as it is derived based on point estimates only. Moreover, the confidence limit of the stability threshold can be considered to be the point at which one might begin to question the extent of incompatibility of the data with the null hypothesis. A final caveat of the proposed confidence limit for the stability threshold is that the confidence interval for the pooled estimate is itself an estimate and subject to error. Nonetheless, if the meta-analysis is accompanied by a high-quality systematic review, this interval estimate is the best available at the time of undertaking the meta-analysis, and this limitation is outweighed by the potential value gained by redirecting research effort and reducing research waste. Finally, it should be stressed that the proposed methods to ascertain statistical sufficiency and stability are intended to complement rather than replace a thorough assessment of causation, uncertainty and heterogeneity.

## 5. Conclusions

Preterm birth is associated with higher exposure to PM_2.5_, and it is unlikely that any new observational study will alter this conclusion. Due to the high statistical heterogeneity, the magnitude of the association remains uncertain. Future epidemiological studies can redirect research efforts away from ascertaining whether an association exists, which is now well-established, to instead characterise this risk (in terms of magnitude, impact, pathways, populations and potential bias) and identify interventions. Sufficiency and stability can be effectively applied in meta-analyses, and these metrics can aid progression towards a consensus epidemiological statement on the association between preterm birth and PM_2.5_ and potentially reduce research waste.

## Figures and Tables

**Figure 1 ijerph-19-02036-f001:**
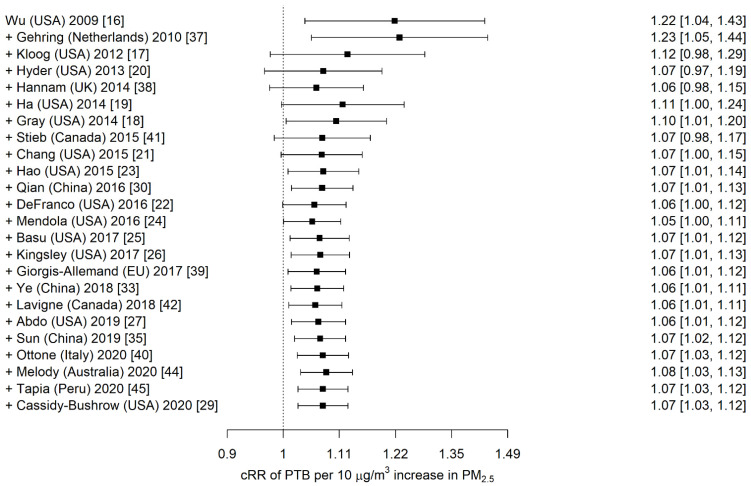
Converging cumulative relative risks (cRR) with each successive publication on the association between preterm birth (PTB) and whole-pregnancy exposure to fine particulate matter (PM_2.5_). The “+” symbol indicates sequential addition of the study results to those previously published.

**Table 1 ijerph-19-02036-t001:** Cumulative relative risks (cRR) of preterm birth (PTB) per 10 µg/m^3^ increase in fine particulate matter (PM_2.5_) and accumulating heterogeneity (I^2^) over time, with the relative risk needed for a new study to render the pooled estimate null (stability threshold, ST).

First Author	Country	Year ^1^	Births (*N*)	PTB (%)	PM_2.5_ Mean (SD) ^2^	RR (95% CI)	cRR (95% CI) ^3^	I^2^ (%)	ST (CI Limit) ^4^
Wu	USA	2009	81,186	8.3	1.8 (1.3)	1.22 (1.07, 1.47)	1.22 (1.04, 1.43)	0	0.81 (1.06)
+Gehring	Netherlands	2010	3853	4.3	20.1 (NA)	1.51 (0.68, 3.23)	1.23 (1.05, 1.44)	0	0.81 (1.06)
+Kloog	USA	2012	634,244	9.8	9.6 (5.1)	1.05 (1.01, 1.12)	1.12 (0.98, 1.29)	51	0.89 (1.08)
+Hyder	USA	2013	656,769	6.3	11.4 (0.8)	0.96 (0.82, 1.12)	1.07 (0.97, 1.19)	48	0.93 (1.07)
+Hannam	UK	2014	38,608	6.5	NA	0.96 (0.72, 1.26)	1.06 (0.98, 1.15)	31	0.94 (1.06)
+Ha	USA	2014	123,207	9.5	9.9 (1.7)	1.26 (1.19, 1.33)	1.11 (1.00, 1.24)	80	0.89 (1.04)
+Gray	USA	2014	457,642	8.9	13.6 (1.7)	1.05 (0.96, 1.09)	1.10 (1.01, 1.20)	80	0.90 (1.03)
+Stieb	Canada	2015	2,966,705	6.2	8.4 (2.4)	0.96 (0.93, 0.99)	1.07 (0.98, 1.17)	89	0.93 (1.05)
+Chang	USA	2015	175,891	10.6	17.0 (NA)	1.07 (1.00, 1.10)	1.07 (1.00, 1.15)	88	0.93 (1.03)
+Hao	USA	2015	511,658	9.2	NA	1.10 (1.03, 1.18)	1.07 (1.01, 1.14)	87	0.93 (1.02)
+Qian	China	2016	95,911	4.5	70.8 (NA)	1.06 (1.04, 1.10)	1.07 (1.01, 1.13)	88	0.93 (1.02)
+DeFranco	USA	2016	224,921	8.5	13.0 (1.6)	0.92 (0.85, 1.00)	1.06 (1.00, 1.12)	90	0.94 (1.02)
+Mendola	USA	2016	223,502	11.7	11.8 (NA)	1.02 (0.98, 1.06)	1.05 (1.00, 1.11)	89	0.94 (1.02)
+Basu	USA	2017	231,637	10	18.8 (4.8)	1.21 (1.18, 1.25)	1.07 (1.01, 1.12)	92	0.93 (1.02)
+Kingsley	USA	2017	61,640	8.1	9.5 (1.5)	1.15 (0.79, 1.65)	1.07 (1.01, 1.13)	91	0.93 (1.02)
+Giorgis-Allemand	EU	2017	46,791	4.9	NA	0.93 (0.77, 1.08)	1.06 (1.01, 1.12)	91	0.94 (1.01)
+Ye	China	2018	24,246	6.2	68.8 (7.8)	1.07 (1.02, 1.13)	1.06 (1.01, 1.11)	90	0.94 (1.01)
+Lavigne	Canada	2018	196,171	7.8	9.0 (2.0)	0.80 (0.53, 1.15)	1.06 (1.01, 1.11)	90	0.94 (1.01)
+Abdo	USA	2019	446,961	14	7.1 (1.6)	1.81 (1.14, 2.68)	1.06 (1.01, 1.12)	90	0.93 (1.02)
+Sun	China	2019	6275	5.9	60.4 (10.8)	1.12 (1.03, 1.23)	1.07 (1.02, 1.12)	89	0.93 (1.02)
+Ottone	Italy	2020	23,708	5.5	18.0 (2.5)	1.32 (1.02, 1.69)	1.07 (1.03, 1.12)	89	0.93 (1.02)
+Melody	Australia	2020	285,594	3	6.9 (NA)	1.34 (1.08, 1.65)	1.08 (1.03, 1.13)	89	0.92 (1.02)
+Tapia	Peru	2020	123,034	7.2	22.3 (5.4)	0.98 (0.95, 1.02)	1.07 (1.03, 1.12)	90	0.93 (1.02)
+Cassidy-Bushrow	USA	2020	7690	10.6	10.7 (1.3)	1.09 (0.56, 2.01)	1.07 (1.03, 1.12)	89	0.93 (1.02)

The “+” symbol indicates sequential addition of the study results to those previously published. NA: not available. CI: confidence interval. SD: standard deviation. ^1^ The earlier of the year available online after acceptance for publication and year of publication. ^2^ Whole-pregnancy mean and standard deviation of PM_2.5_ when reported, and study period mean and standard deviation when not reported. ^3^ Pooled relative risk for all studies up to and including the study specified in the row. ^4^ The relative risk for a new study to render the closest CI limit of the pooled RR to 1 after including the new study.

**Table 2 ijerph-19-02036-t002:** Cumulative hazard ratios (cHR) of preterm birth (PTB) per 10 µg/m^3^ increase in fine particulate matter (PM_2.5_) and accumulating heterogeneity (I^2^) over time, with the hazard ratio needed for a new study to render the pooled estimate null (stability threshold, ST).

First Author	Country	Year ^1^	Births (*N*)	PTB (%)	PM_2.5_ Mean (SD) ^2^	HR (95% CI)	cHR (95% CI) ^3^	I^2^ (%)	ST (CI Limit) ^4^
Chen	Australia	2017	173,720	7.7	6.2 (NA)	1.45 (1.16, 1.79)	1.45 (1.17, 1.80)	0	0.69 (1.12)
+Wang	China	2018	469,975	5.5	39.1 (22.7)	1.00 (0.81, 1.23)	1.20 (0.84, 1.73)	83	0.83 (1.20)
+Guo	China	2018	426,246	8.3	63.4 (24.9)	1.06 (1.05, 1.06)	1.14 (0.93, 1.40)	81	0.87 (1.13)
+Li	China	2018	1,240,978	8.1	53.4 (15.9)	1.09 (1.08, 1.10)	1.10 (1.01, 1.21)	99	0.90 (1.03)
+Yuan	China	2019	3692	4.6	49.3 (5.0)	0.92 (0.60, 1.41)	1.08 (1.05, 1.11)	90	0.92 (1.02)
+Sheridan	USA	2019	2,293,218	8.2	13.5 (NA)	1.12 (1.09, 1.14)	1.09 (1.06, 1.13)	93	0.91 (1.02)
+Liang	China	2019	628,439	4.7	36.9 (NA)	0.98 (0.93, 1.02)	1.08 (1.01, 1.14)	98	0.92 (1.02)

NA: not available. CI: confidence interval. SD: standard deviation. ^1^ The earlier of the year available online after acceptance for publication and year of publication. ^2^ Whole-pregnancy mean and standard deviation of PM_2.5_ when reported, and study period mean and standard deviation when not reported. ^3^ Pooled hazard ratio for all studies up to and including the study specified in the row. ^4^ The relative risk for a new study to render the closest CI limit of the pooled RR to 1 after including the new study.

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
