# Peer review of "A Simple Method to Establish Sufficiency and Stability in Meta-Analyses: With Application to Fine Particulate Matter Air Pollution and Preterm Birth"

_ijerph, 2022, doi:10.3390/ijerph19042036_

Round 1

Reviewer 1 Report

The authors proposed a new method to identify whether a new epidemiological study is needed to ascertain whether an observational association exists between PM2.5 and preterm birth. This study conclude that preterm birth is associated with PM2.5 and it is unlikely that any new observational study will alter this conclusion.

1. The methodology appears to be adequate but it would be better if a reference is provided.

2. Please compare the pooled estimates with recent meta-analyses and discuss the difference if it exists.

3. It would be better to analyze or discuss the possible heterogeneity among various different epidemiological studies.

4. It may be arbitrary to say that it is unlikely that any new observational study will alter this conclusion.

Reviewer 2 Report

This study aimed to develop a method that could establish a stable estimate of associations in a meta-analysis. 

Thank you for this well structured and interesting manuscript. I have a few basic comments:

  1. Throughout the manuscript you use the term "Fine particulate matter air pollution" which is not necessary. I would suggest using "fine particulate matter" as it is a standard term within the field of environmental epidemiology.
  2. Please clarify in the abstract that you investigated the association of preterm birth and exposure of PM2.5 throughout the whole pregnancy.
  3. In Line 44 you state "... establishment of plausible biological pathways..." however, you do not state what these pathways are. I would give an example of these potential pathways.
  4. In the method section (specifically section 2.2) you tell us the included studies were selected from a recent systematic review and meta-analysis. I would suggest giving a bit more detail on this systematic review considering that your study is based on it.
  5. In the methods section (section 2.3) you give a definition of what preterm birth is. Please give a reference for this definition.
  6. In the methods section (section 2.4) please clarify whether you did the data extraction yourself, or if the data extraction was taken from the systematic review. Also please confirm if the data was extracted in duplicate.
  7. In the methods section (section 2.6). Line 129-130 should be revised.
  8.  I would suggest that you clarify in the methods section that you used I2 to quantify heterogeneity and additionally could you clarify the scale and definition of high heterogeneity that you used.
  9.  I feel it is important to consider that the WHO reports that 60% of premature births occur in two specific regions: sub-Saharan Africa and South Asia; however, your study doesn't included a single study from these regions. This makes drawing an actual conclusion difficult and as such I would suggest mentioning it in your study limitations sub section.
  10. In the conclusion please give the direction of the effect between preterm births and PM2.5 exposure.

Reviewer 3 Report

The reviewer appreciates the author for integrating an interesting approach in the meta-analysis. The comments are attached

Round 2

Reviewer 2 Report

I thank the authors for their revision and have no further comments.

Reviewer 3 Report

The reviewer is satisfied with the corrections made by the author.